# Identification of an emphysema-associated genetic variant near *TGFB2* with regulatory effects in lung fibroblasts

Margaret M Parker[1], Yuan Hao[1], Feng Guo[1], Betty Pham[1], Robert Chase[1], John Platig[1], Michael H Cho[1,2], Craig P Hersh[1,2], Victor J Thannickal[3], James Crapo[4], George Washko[2], Scott H Randell[5], Edwin K Silverman[1,2], Raúl San José Estépar[6], Xiaobo Zhou[1]*, Peter J Castaldi[1,7]*

[1]Channing Division of Network Medicine, Brigham and Women's Hospital, Boston, United States; [2]Division of Pulmonary and Critical Care Medicine, Brigham and Women's Hospital, Boston, United States; [3]Division of Pulmonary, Allergy and Critical Care, Department of Medicine, School of Medicine, University of Alabama at Birmingham, Birmingham, United States; [4]Division of Pulmonary, Critical Care and Sleep Medicine, National Jewish Health, Denver, United States; [5]Marsico Lung Institute, The University of North Carolina at Chapel Hill, Chapel Hill, United States; [6]Applied Chest Imaging Laboratory, Brigham and Women's Hospital, Boston, United States; [7]Division of General Internal Medicine and Primary Care, Brigham and Women's Hospital, Boston, United States

*For correspondence:
xiaobo.zhou@channing.harvard.edu (XZ);
peter.castaldi@channing.harvard.edu (PJC)

**Abstract** Murine studies have linked TGF-β signaling to emphysema, and human genome-wide association studies (GWAS) studies of lung function and COPD have identified associated regions near genes in the TGF-β superfamily. However, the functional regulatory mechanisms at these loci have not been identified. We performed the largest GWAS of emphysema patterns to date, identifying 10 GWAS loci including an association peak spanning a 200 kb region downstream from *TGFB2*. Integrative analysis of publicly available eQTL, DNaseI, and chromatin conformation data identified a putative functional variant, rs1690789, that may regulate *TGFB2* expression in human fibroblasts. Using chromatin conformation capture, we confirmed that the region containing rs1690789 contacts the *TGFB2* promoter in fibroblasts, and CRISPR/Cas-9 targeted deletion of a ~ 100 bp region containing rs1690789 resulted in decreased *TGFB2* expression in primary human lung fibroblasts. These data provide novel mechanistic evidence linking genetic variation affecting the TGF-β pathway to emphysema in humans.
DOI: https://doi.org/10.7554/eLife.42720.001

## Introduction

Emphysema, that is pathologic destruction of lung parenchyma resulting in airspace enlargement, is one of the major manifestations of chronic obstructive pulmonary disease (COPD). Emphysema occurs in distinct pathologic patterns, but these patterns are not captured by traditional quantitative measures of emphysema from lung computed tomography (CT). In order to have more detailed radiographic measures of emphysema, we developed novel image extraction techniques to quantify the distinct patterns of emphysema based on the analysis of local lung density histograms (*Mendoza, 2012*). These local histogram emphysema (LHE) measures are more predictive of clinical outcomes than standard CT emphysema quantifications (*Castaldi et al., 2013*), and in a previous genome-wide association study (GWAS) we identified genome-wide significant

**eLife digest** It is well known that smoking is bad for the lungs. Not only can smoking cause lung cancer, it can also lead to conditions such as emphysema. This is the gradual damage to lung tissue that occurs when the walls of the tiny air-sacs in the lungs where the blood takes up oxygen, called the alveoli, weaken and break. Emphysema causes shortness of breath and difficulty pushing air out of the lungs, and it is part of chronic obstructive pulmonary disease (also known as COPD).

Genetic differences mean that certain people are more likely to develop emphysema than others. As an example, if someone has genetic mutations that alter the activity of a gene called *TGFB2*, their risk of developing emphysema increases. However, the specific genetic mutations that modify the activity of *TGFB2* were previously unknown.

Parker et al. analyzed the genetic sequences of *TGFB2* from patients with emphysema and compared them to those from healthy individuals. This revealed that certain mutations near the *TGFB2* gene were more common in patients with emphysema. Next, Parker et al. showed that, in healthy lung cells called fibroblasts, the stretch of DNA that was mutated in patients with emphysema touched the part of *TGFB2* that controls when the gene is activated. Deleting that same stretch of DNA in the fibroblasts meant the cells could no longer activate the *TGFB2* gene as efficiently. Together, these results reveal a genetic difference that increases the risk for emphysema.

COPD affects approximately 175 million people worldwide, causing over three million deaths each year. The findings of Parker et al. suggest that developing drugs that safely and efficiently target *TGFB2* may be a way to help patients with early signs of emphysema.
DOI: https://doi.org/10.7554/eLife.42720.002

associations with these distinct LHE patterns (*Castaldi et al., 2014*). However, the mechanisms by which these GWAS loci affect emphysema patterns are unknown.

The majority of GWAS-identified loci for genetically complex diseases are located in non-coding DNA and influence gene regulatory elements (*Maurano et al., 2012*; *Nicolae et al., 2010*). Thus, for the functional characterization of emphysema GWAS loci, it is necessary to localize causal variants in regulatory elements and identify the gene(s) regulated by that element. Since multiple cell types contribute to emphysema, large-scale functional annotation projects such as the Genotype-Tissue Expression Project (GTEx) (*GTEx Consortium, 2015*) and the Encyclopedia of Regulatory Elements (ENCODE) (*ENCODE Project Consortium, 2012*) can be integrated with GWAS signals to identify candidate regulatory regions, tissues, and cell types of interest for more detailed functional characterization.

In this study, we hypothesized that human emphysema is influenced by functional genetic variants that disrupt gene regulatory elements. As a screening approach, we cross-referenced GWAS results against large compendia of gene regulatory data from tissues and cell types to prioritize emphysema-associated loci for further functional study. This analysis identified rs1690789 as a high-probability functional variant in the GWAS-identified region near *TGFB2*. Using chromatin conformation capture, we confirmed that the region spanning this SNP interacts with the *TGFB2* promoter region. Via CRISPR/Cas-9 targeted deletion, we then demonstrated that a ~ 100 bp segment containing rs1690789 increases *TGFB2* expression in primary human lung fibroblasts, providing novel evidence that genetic variation affecting TGF-β signaling contributes to the genetic predisposition to emphysema.

## Results

### Validation of LHE clinical and genetic associations

In subjects from the COPDGene Study, we have previously demonstrated that LHE measures are associated with COPD-related phenotypes (*Castaldi et al., 2013*) and with common genetic variants at genome-wide significance (*Castaldi et al., 2014*). To confirm these associations in an independent cohort and discover new genetic associations, we generated new LHE measures in 1519 subjects from the ECLIPSE Study, and we replicated the previously observed relationships between LHE pattern and GOLD (Global Initiative for Obstructive Lung Disease) spirometric grade (*Figure 1*). In the

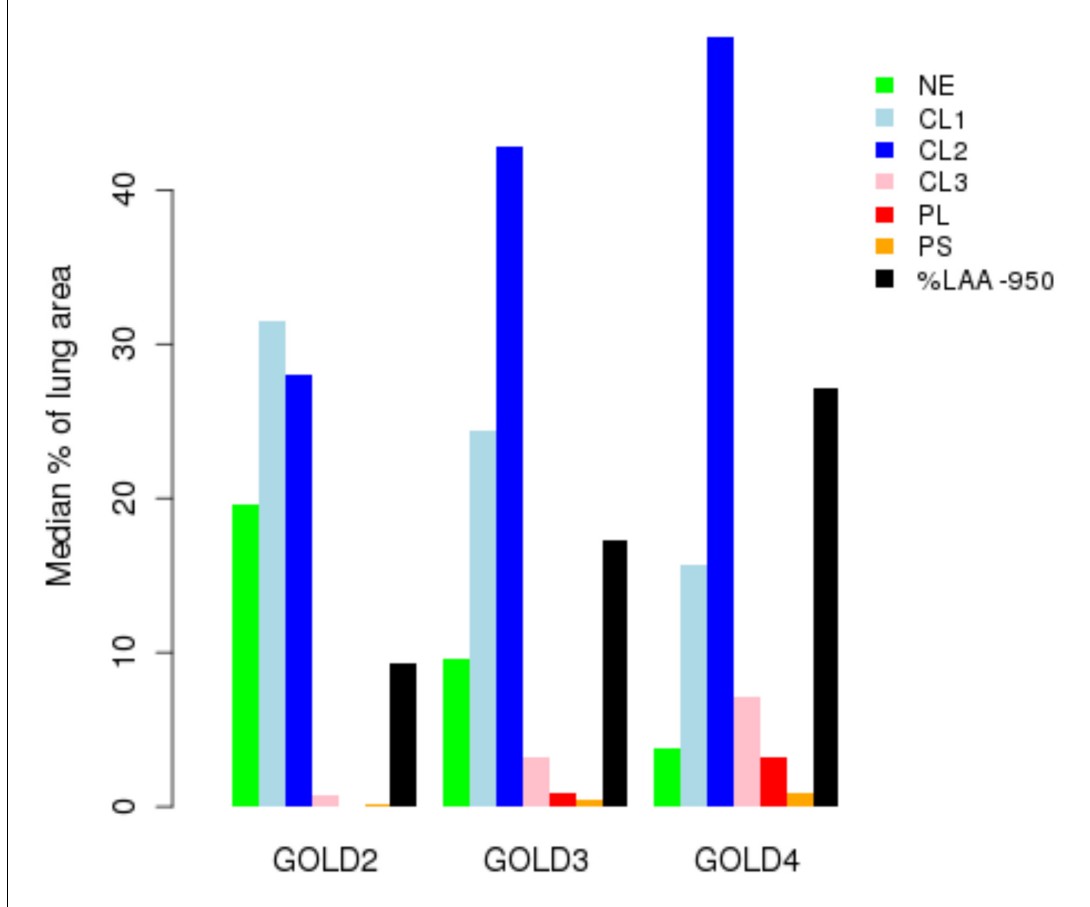

**Figure 1.** Percentage of each LHE-based emphysema pattern by Global Obstructive Lung Disease (GOLD) stage in ECLIPSE.  NE = Non-emphysematous lung. CL1 = Mild centrilobular. CL2 = Moderate centrilobular. CL3 = Severe centrilobular. PL = Panlobular. PS = Paraseptal. %LAA-950 = emphysema based on −950 Hounsfield unit threshold.
DOI: https://doi.org/10.7554/eLife.42720.003

combined GWAS meta-analysis, we identified 10 independent regions with genome-wide significant associations to at least one LHE phenotype, six of which had been previously described (*Table 1*). One of the four novel associations is rs28929474, the pathogenic Glu→Lys substitution in SERPINA1 which is known to be associated with COPD. There was no evidence of systematic inflation in the QQ-plots of these GWAS (*Figure 2*). Subject characteristics are shown in *Supplementary file 1* Table 1, and complete results by cohort are shown in *Supplementary file 1* Table 2.

Since the more severe emphysema patterns (severe centrilobular and panlobular emphysema) are non-normally distributed, we performed a sensitivity analysis for these top results after performing inverse normal transformation of the LHE pattern phenotypes (*Supplementary file 1* Table 3). In this analysis, four loci remained genome-wide significant (loci on chromosome 15, 14, 11, and 1), two loci had p-values$<5\times10^{-7}$, and four associations to the panlobular and severe centrilobular patterns had notably lower p-values suggesting that these specific associations are driven by extreme phenotype values and should be interpreted with caution.

With regard to the association with the SERPINA1 Z-allele (rs28929474), subjects with known alpha-1 antitrypsin deficiency had been excluded from our primary analysis. However, when we examined the imputed genotypes of rs28929474, we identified six individuals in ECLIPSE with imputed PiZZ genotypes. When we repeated the genetic analysis without these subjects, there was an increase in association p-value in ECLIPSE (0.003 versus 0.0004, consistent direction of effect), and the meta-analysis association p-value was $1.6 \times 10^{-7}$.

**Table 1.** Genome-wide significant meta-analysis results for local histogram emphysema phenotypes.

| LHE pattern | Lead SNP | Chromosome | Position (hg19) | EAF | Effect allele | P value meta | Effect meta | SE effect meta |
|---|---|---|---|---|---|---|---|---|
| Moderate Centrilobular | rs56077333 | 15 | 78899003 | 0.67 | a | 2.2E-13 | 0.016 | 2.2E-03 |
| | rs17368582 | 11 | 102738075 | 0.14 | t | 8.1E-12 | 0.024 | 3.5E-03 |
| | rs56113850 | 19 | 41353107 | 0.59 | t | 1.5E-09 | −0.016 | 2.6E-03 |
| | rs796395 | 1 | 218681971 | 0.52 | a | 6.1E-09 | 0.013 | 2.2E-03 |
| | rs138641402 | 4 | 145445779 | 0.38 | a | 3.3E-08 | 0.014 | 2.5E-03 |
| Nonemphysematous (Normal Lung) | rs138641402 | 4 | 145445779 | 0.38 | a | 4.2E-08 | −0.021 | 3.9E-03 |
| | rs7170068 | 15 | 78912943 | 0.78 | a | 4.8E-12 | 0.028 | 4.0E-03 |
| | rs17368659 | 11 | 102742761 | 0.86 | t | 6.3E-11 | 0.036 | 5.4E-03 |
| | rs28929474* | 14 | 94844947 | 0.98 | t | 6.2E-09 | −0.071 | 1.2E-02 |
| Panlobular | rs11852372 | 15 | 78801394 | 0.35 | a | 7.1E-12 | −0.003 | 5.0E-04 |
| | rs76756075* | 11 | 112349844 | 0.02 | t | 2.0E-08 | −0.007 | 1.2E-03 |
| | rs78070126* | 11 | 6574608 | 0.97 | t | 2.8E-08 | 0.006 | 1.1E-03 |
| | rs145770770 | 2 | 152487808 | 0.99 | a | 3.8E-08 | −0.015 | 2.8E-03 |
| Severe Centrilobular | rs9788721 | 15 | 78802869 | 0.38 | t | 3.4E-15 | −0.005 | 6.0E-04 |
| | rs379123 | 17 | 30891814 | 0.40 | t | 3.7E-08 | −0.004 | 7.0E-04 |

LHE - local histogram emphysema.

EAF - effect allele frequency in 1000 Genomes CEU population.

*indicates novel association not previously associated in GWAS of COPD or emphysema (rs28929474 was associated to FEV1/FVC in smokers in **Li et al., 2018**, during preparation of this manuscript).

DOI: https://doi.org/10.7554/eLife.42720.005

To determine whether these variants were associated with other COPD-related phenotypes, we queried the LHE GWAS significant associations against the results from two recent large GWAS studies for $FEV_1$, $FEV_1/FVC$, and COPD status (**Shrine et al., 2019**; **Sakornsakolpat et al., 2019**). Five of the 10 LHE loci (lead variants rs56113850, rs796395, rs17368659, rs145770770, and the 15q25 locus) were associated to at least one of these outcomes at p<0.05 with a consistent direction of effect (**Supplementary file 1** Table 4).

Some loci associated with COPD and related phenotypes have also been associated with smoking behavior, raising the question of whether the COPD associations at these loci are mediated through smoking. To determine how many of our associations were also associated to smoking behavior, we queried our results against the UK Biobank Pheweb server GWAS for prior history of smoking, and we observed that the only associations that were nominally associated to smoking were the previously known smoking associations in the 15q25 and 19q13 loci (**Supplementary file 1** Table 5).

## eQTL colocalization analysis to identify candidate GWAS target genes and tissue enrichment of LHE GWAS signals

To generate functional hypotheses for emphysema-associated loci and prioritize regions for further functional study, we integrated our GWAS results with large-scale genome-wide eQTL and cell type epigenomic data, as shown in **Figure 3**. To identify emphysema-associated loci that overlap with eQTL signals from multiple tissues, we cross-referenced our LHE GWAS results against eQTL results from 44 GTEx tissues and blood eQTLs from COPDGene. Since overlap between GWAS and eQTL signals can be due to chance, we used a Bayesian colocalization method (**Giambartolomei et al., 2014**) to quantify the probability that the local GWAS and eQTL signals were attributable to a shared causal variant. Four genome-wide significant LHE regions overlapped with eQTL regions with an estimated >80% probability of a shared causal variant responsible for the GWAS and eQTL associations (**Table 2**).

To identify additional candidate colocalization loci that may be present below the stringent genome-wide significance threshold, we studied SNPs with a GWAS p<5×10$^{-5}$. At this threshold, the number of GWAS-eQTL overlap loci ranged from 78 (panlobular pattern) to 159 (moderate

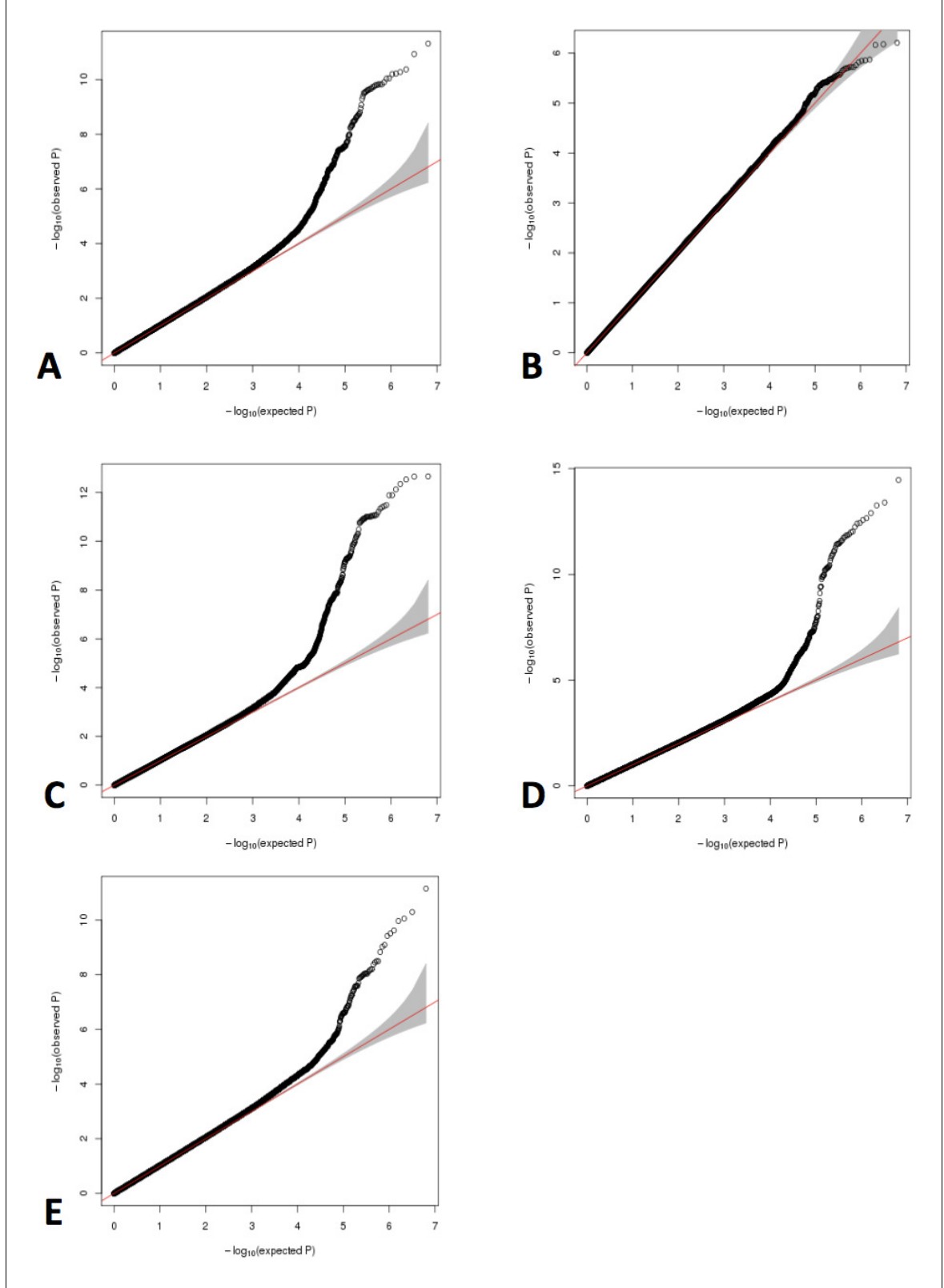

**Figure 2.** QQ plots for GWAS meta-analyses of non-transformed LHE phenotypes. (**A**) Nonemphysematous lung. (**B**) Mild centrilobular pattern. (**C**) Moderate centrilobular. (**D**) Severe centrilobular. (**E**) Panlobular.

DOI: https://doi.org/10.7554/eLife.42720.004

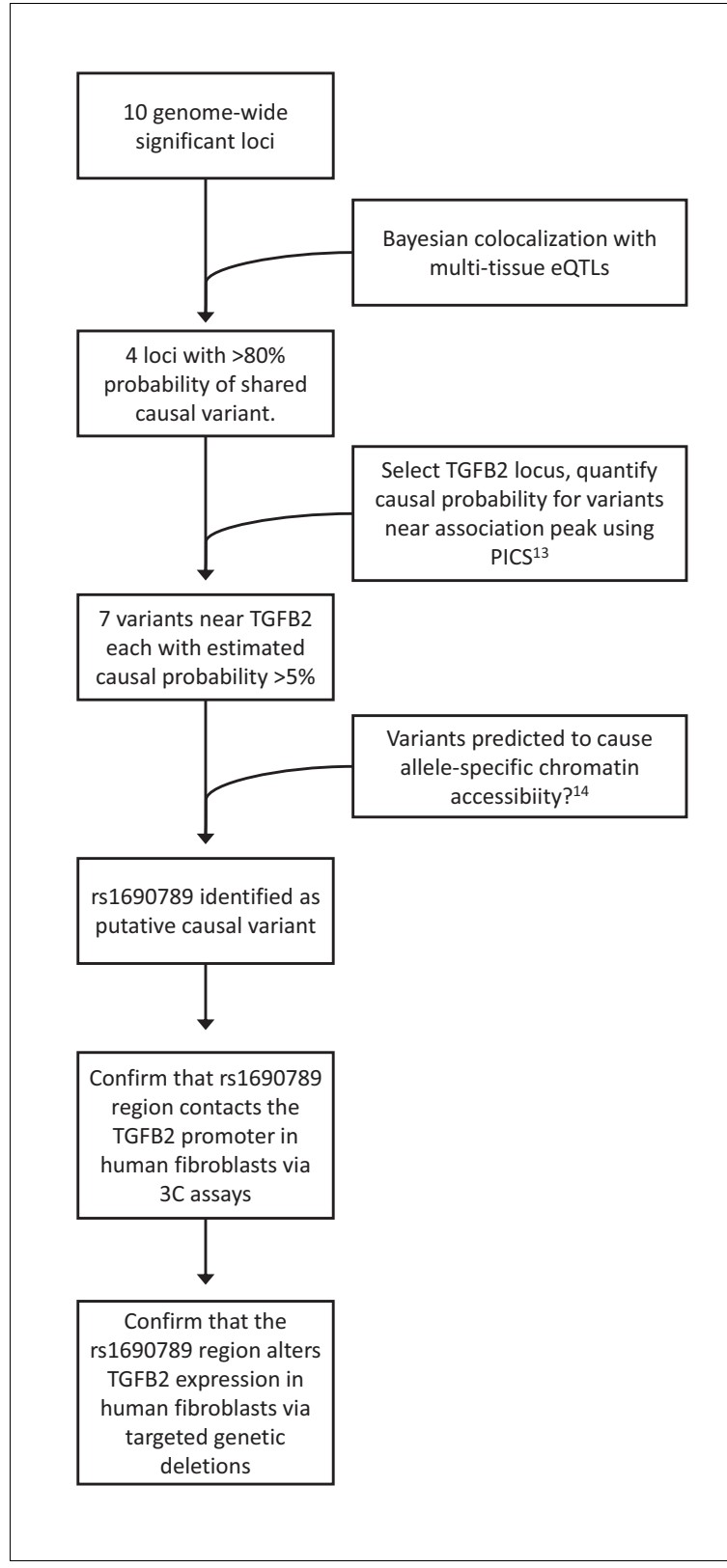

**Figure 3.** Overview of integrative analyses to prioritize genome-wide significant emphysema-associated loci for functional studies.

DOI: https://doi.org/10.7554/eLife.42720.006

**Table 2.** Genomewide significant LHE loci that colocalize with eQTL.

| GWAS | SNP | CHR | POS | Tissue | GENE |
|------|-----|-----|-----|--------|------|
| Moderate centrilobular | rs796395 | 1 | 218681971 | Fibroblasts | *TGFB2* |
| | rs56077333 | 15 | 78899003 | Fibroblasts, Testis | *PSMA4, CHRNA5* |
| Moderate centrilobular, normal | rs56113850 | 19 | 41353107 | Lung, Testis | *CYP2A6, AKT2* |
| Panlobular | rs11852372 | 15 | 78801394 | Testis | *CHRNA5* |

All loci have colocalization probability > 80%, reflecting the estimated probability that the GWAS and eQTL association signals arise from a shared causal variant.
LHE pattern – local histogram emphysema phenotype used for GWAS.
SNP - lead GWAS variant in locus.
Tissue - tissue of origin for gene expression data in eQTL analysis.
Gene – eQTL targeted gene in specified GTEx tissue.

DOI: https://doi.org/10.7554/eLife.42720.007

centrilobular pattern), representing between 15% to 33% of the total number of loci with a GWAS $p<5\times10^{-5}$. Of these loci, 32 had a > 80% estimated probability of having a shared causal GWAS-eQTL variant, and we identified the genes whose expression levels are altered by these loci (*Supplementary file 1* Table 6). Full results of this analysis are available at https://cdnm.shinyapps.io/lhemphysema_eqtlcolocalization/.

To test for tissue-specific enrichment of LHE GWAS signals, we quantified the enrichment of LHE GWAS regions associated at $p<5\times10^{-5}$ in DNaseI peak regions from ENCODE and Roadmap cell types using the Garfield method (*Iotchkova et al., 2019*). The most commonly enriched cell types were fibroblasts and fetal lung tissue, as can be seen in the enrichment results for moderate centrilobular emphysema (*Figure 4*). Out of 424 tested cell type annotations, there were 15, 25, and 1 cell type that exceeded the significance threshold for the moderate centrilobular, nonemphysematous, and severe centrilobular LHE phenotypes, respectively (*Supplementary file 1* Table 7).

## Fine mapping identifies a candidate causal variant in the TGFB2 locus

One of the top GWAS-eQTL colocalization signals associated with the moderate centrilobular emphysema pattern spans a 200 kb region that includes the 3' UTR of *TGFB2* and extends 100 kb downstream. Multiple SNPs in this region were significantly associated with *TGFB2* expression in human tissues from the GTEx project (*Figure 5*) with the highest colocalization present with the eQTL signal in cultured fibroblasts. Given the essential roles of TGF-β signaling and fibroblasts in lung repair pathways, we selected this locus for further investigation.

To confirm the colocalization results for *TGFB2*, we performed a separate colocalization analysis using the same eQTL data but a separate colocalization methodology (*He et al., 2013*). Sherlock analysis for the moderate centrilobular GWAS results and GTEx eQTL data from fibroblasts, lung tissue, and whole blood confirmed *TGFB2* as a colocalization target for moderate centrilobular emphysema in fibroblasts, and a total of nine colocalizing genes or transcripts were identified at a p-value$<1\times10^{-4}$ (*Supplementary file 1* Table 8).

The GWAS signal in this region appears to demonstrate two independent peaks of association spanning a recombination hotspot, with the fibroblast eQTL signal appearing to colocalize with only one of these signals. We performed conditional genetic association analysis of this region, confirming the presence of two independent signals (secondary association lead SNP rs3009942 $p=4.4\times10^{-7}$, *Figure 6*). To confirm that these are independent signals, we also performed conditional association adjusting for rs3009942, which minimally attenuated the primary association (rs796395 conditional p-value$=3.3\times10^{-7}$).

Focusing on the primary association peak which colocalized with the fibroblast eQTL signal, we estimated the causal probability (i.e. the likelihood that each individual SNP is the causal variant) of each SNP in this region using the PICS method (*Farh et al., 2015*), identifying seven variants each with a > 5% estimated likelihood to be causal (*Supplementary file 1* Table 9). We then queried whether any of these seven SNPs were predicted to alter transcription factor occupancy using the results of a previously published model developed from ENCODE data (*Maurano et al., 2015*), identifying rs1690789 (minor allele frequency of 0.48 in 1000 Genomes EUR population) as the only variant in this set predicted to have allele-specific effects on transcription factor occupancy.

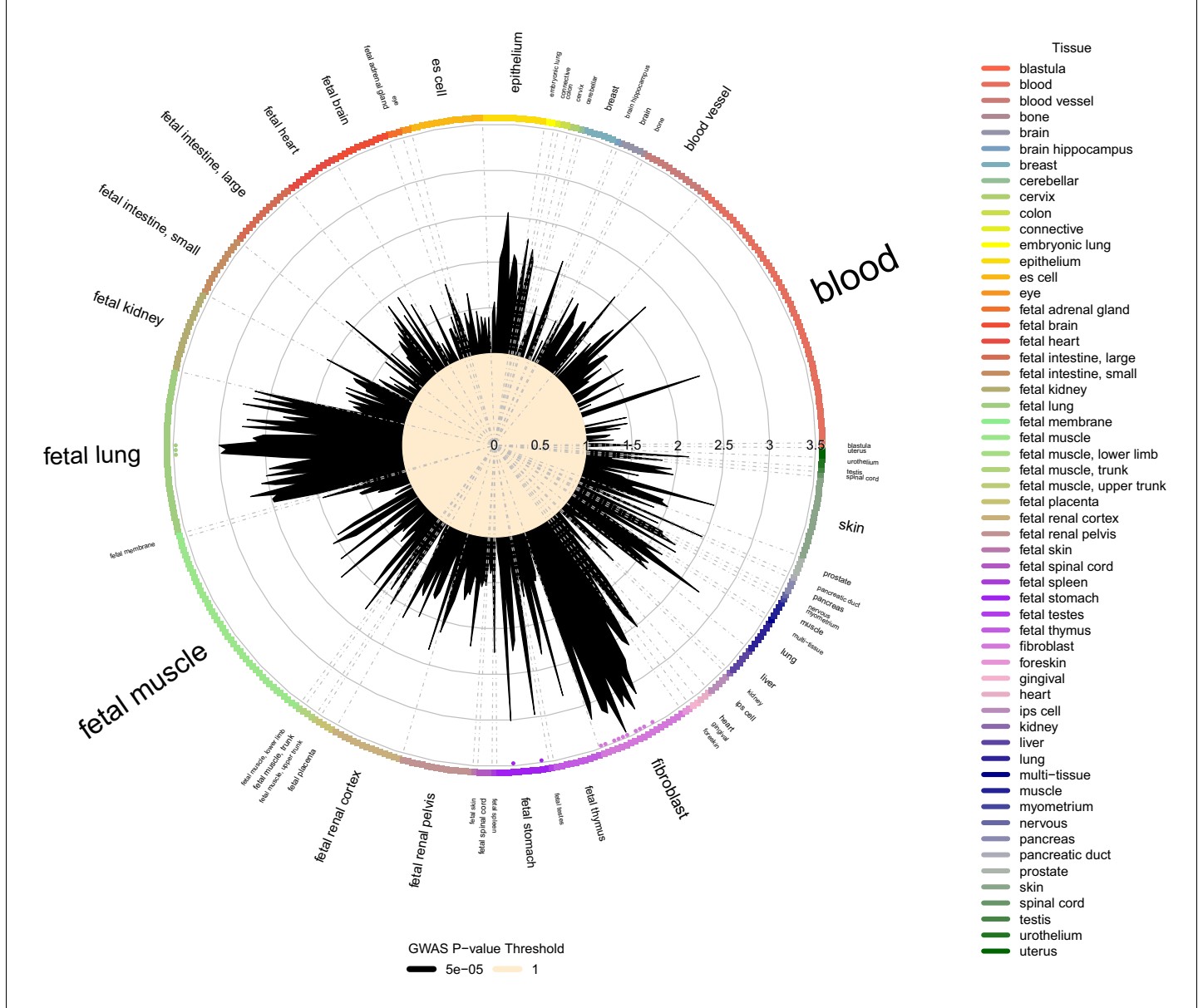

**Figure 4.** Cell type and tissue enrichment for moderate centribloular emphysema GWAS signals. Using Garfield (*Iotchkova et al., 2019*) for enrichment analysis, we tested the enrichment of moderate centrilobular GWAS loci (harboring associations a $p<5\times10^{-5}$) in DNaseI peaks from 424 cell lines and cell types in ENCODE and Roadmap. Significant enrichments were observed in fetal lung, fetal stomach, and multiple fibroblast cell types. These significant enrichments are denoted by colored dots located just inside the boundary of the circle of cell types.

DOI: https://doi.org/10.7554/eLife.42720.008

## Analysis of DNaseI accessibility near rs1690789 across various cell types in publicly available data

Using the ENCODE uniformly processed DNaseI hypersensitivity dataset of 125 cell types, we observed that rs1690789 lies within a DNaseI hypersensitivity peak identified in 13 cell types (*Figure 5*, Panel C). Eight of these 13 cell types were fibroblasts, although this peak was not universally detected in all fibroblast DNaseI experiments, suggesting that this may be a context-specific regulatory element or that DNaseI accessibility may be influenced by genetic variation in these cell types.

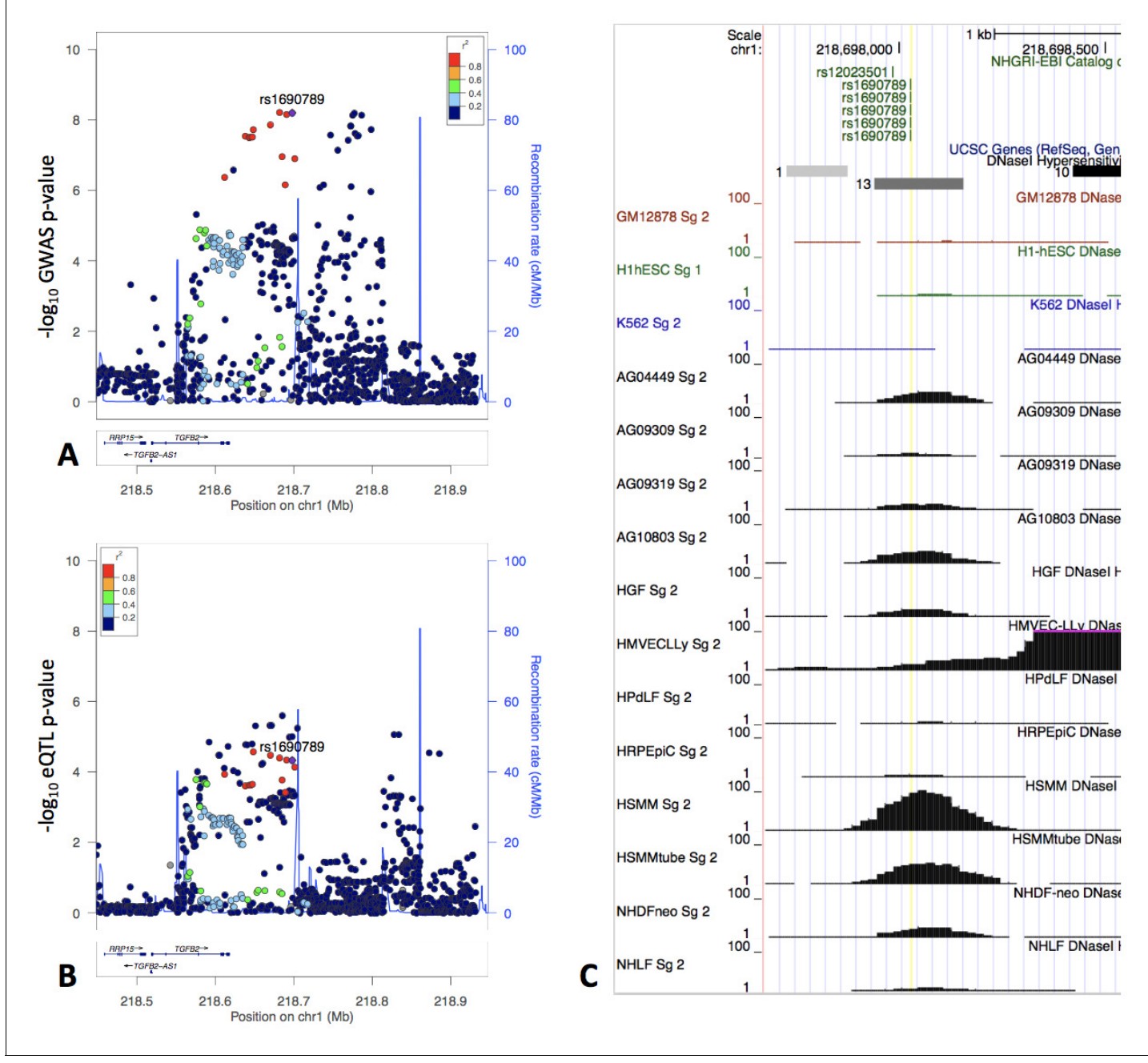

**Figure 5.** The locus zoom plot of GWAS p-values suggests two independent associations (Panel A), and the GWAS signal colocalizes with an eQTL signal in fibroblasts from GTEx (Panel B). rs1690789 is located at one of these GWAS association peaks and lies within a context-specific DNaseI peak (Panel C). GM12878, H1hESC, and K562 cell lines are shown for reference, and the remaining cell types are those with DNaseI peaks that overlap rs1690789. Raw DNaseI data from only one experimental replicate are shown. GM12878 = lymphblastoid cell line. H1hESC = human embryonic stem cell. K562 = leukemia cell line. AG0449-AG10803 refer to fibroblasts from different subjects and sampling sites. HGF = gingival fibroblasts. HMVECLLy = lung derived microvascular endothelial cells. HPdLF - Periodontal ligament fibroblasts. HRPEpiC – retinal pigment epithelial cells. HSMM – skeletal muscle myoblasts. NHDF – dermal fibroblasts. NHLF – lung fibroblasts.
DOI: https://doi.org/10.7554/eLife.42720.009

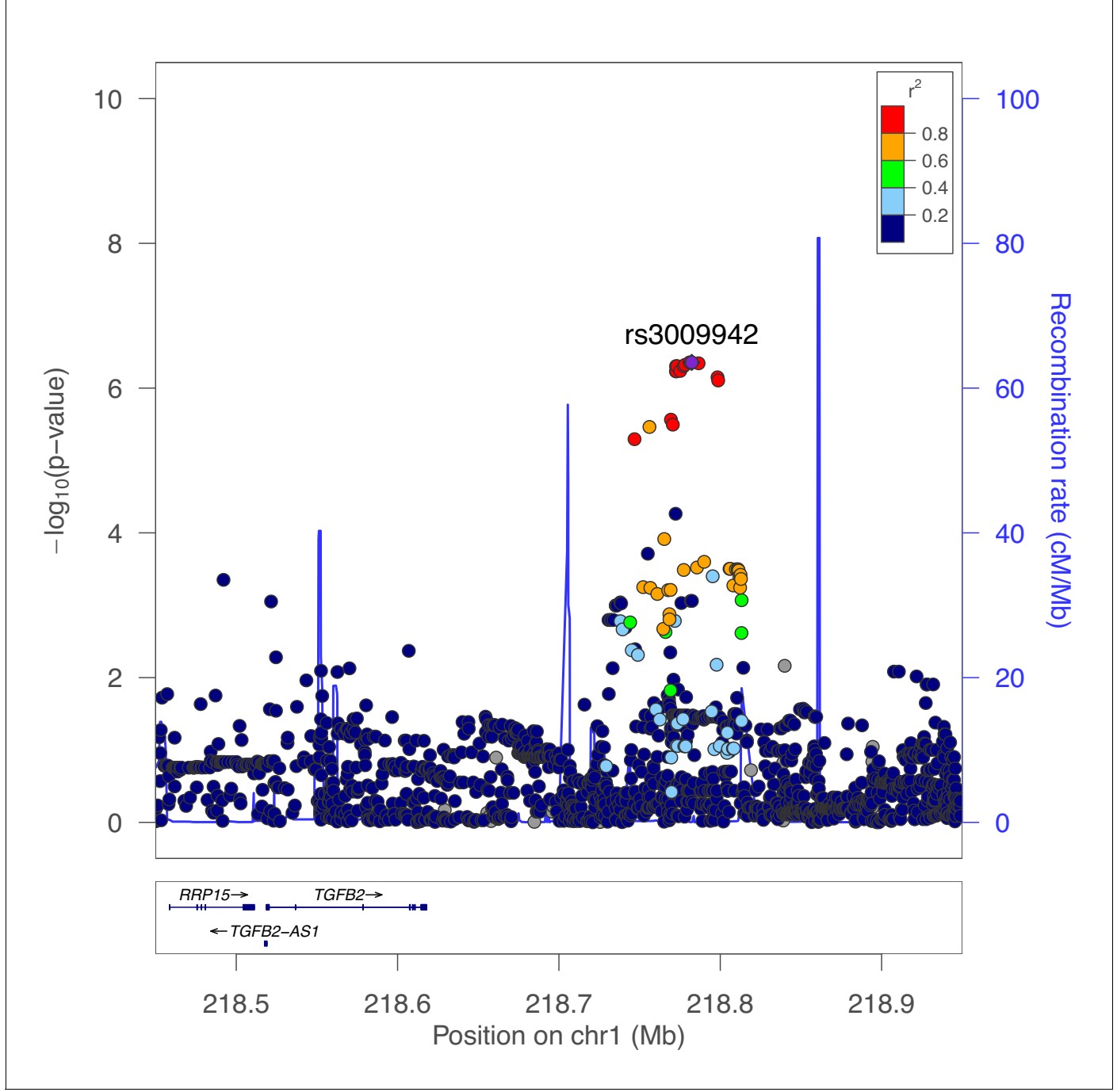

**Figure 6.** Secondary association for moderate centrilobular emphysema at the TGFB2 locus. Results from genetic association meta-analysis conditioned on the lead SNP (rs796395) at this region.

DOI: https://doi.org/10.7554/eLife.42720.010

## Chromatin interaction between GWAS peak regions and the TGFB2 promoter

Since rs1690789 is located ~200 kb from the transcription start site of *TGFB2*, we hypothesized that this region may regulate *TGFB2* expression via a long-range chromatin interaction. Using publicly available 4C-Seq chromatin conformation data from IMR90 human lung fibroblasts (*Rao et al., 2014*), we observed that the 10 kb region containing rs1690789 contacts multiple upstream and

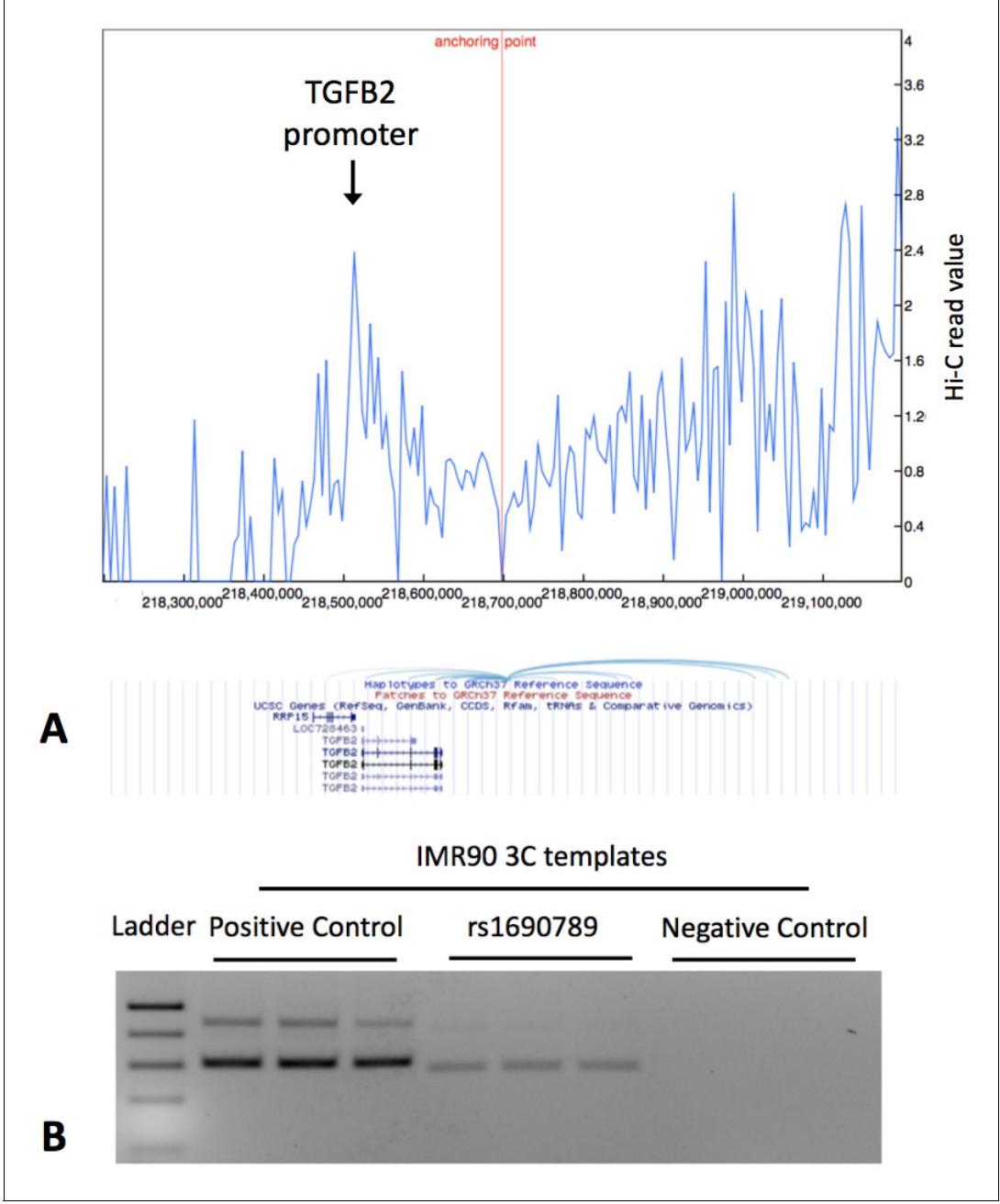

**Figure 7.** Publicly available chromatin conformation capture (4C) results in IMR90 cells show multiple peaks of interaction for the 10 kb region containing the context-specific regulatory element around rs1690789 (Panel A – blue spikes in top figure indicate regions of high interaction frequency and light blue curved lines in lower figure indicate chromosomal interactions with the 10 kb region containing rs1690789). Newly generated 3C assays in IMR90 fibroblasts verify the interaction between the region containing rs1690789 and *TGFB2* promoter (Panel B). Primer sequences are listed in *Supplementary file 1* Table 10.

DOI: https://doi.org/10.7554/eLife.42720.011

downstream regions around *TGFB2* (*Figure 7A*), suggesting that this region is a hotspot of chromosomal interaction.

To confirm whether rs1690789 region indeed interacts with the promoter of *TGFB2* in lung fibroblasts, we performed chromatin conformation capture (3C) experiments in human lung fibroblasts (IMR90). Using the *TGFB2* promoter region as the anchor region, we detected interaction between

the rs1690789-containing region and the *TGFB2* promoter in lung fibroblasts (*Figure 7B*), suggesting long range regulation of *TGFB2* by the region containing rs1690789.

## Deletion of the rs1690789 region alters TGFB2 expression in lung fibroblasts

To determine whether the DNA region near rs1690789 has regulatory effects on the expression of *TGFB2* in human lung fibroblasts in the endogenous genomic context, we generated CRISPR/Cas-9 constructs containing gRNA pairs targeting the ~100 bp region spanning rs1690789 (*Figure 8A*) to generate genomic deletions in normal primary human lung fibroblasts. With sufficient deletion efficiency of the region spanning rs1690789, we detected reduced expression of *TGFB2* (*Figure 8B and C*, *Supplementary file 1* Table 11), indicating that this distal genomic region has regulatory effects on the expression of *TGFB2* in normal primary human lung fibroblasts.

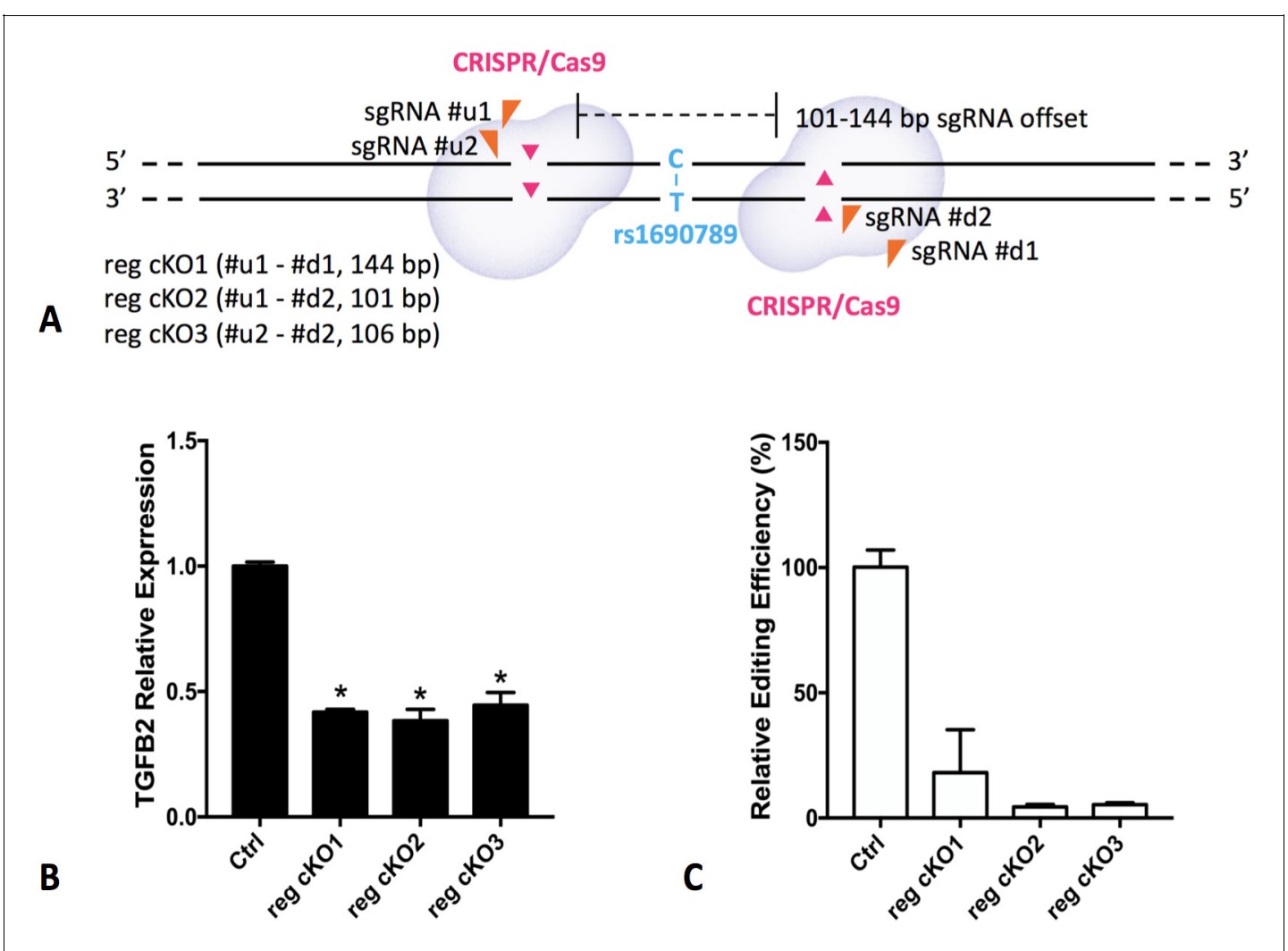

**Figure 8.** Regional knockout of rs1690789 in primary human lung fibroblasts using CRISPR/Cas9 editing. Three pairs of sgRNAs were applied to delete a DNA region of ~100 bp spanning rs1690789 (A). The expression of TGFB2 is downregulated in rs1690789 knockout lung fibroblast cells with qPCR quantification, n = 4 (B). The editing efficiency is examined to confirm the effect of CRISPR/Cas9 regional knockout, n = 4 (C). *p<0.05 compared to control by unpaired *t* test.

DOI: https://doi.org/10.7554/eLife.42720.012

## Discussion

Previous GWAS studies have demonstrated that common genetic variation contributes to emphysema (*Cho et al., 2015*), likely through the perturbation of gene regulatory mechanisms (*Castaldi et al., 2014*). In order to identify putative causal variants and regulatory mechanisms for these loci, we used a screening approach that leverages large compendia of gene regulatory information in the GTEx and ENCODE projects. Using Bayesian colocalization, we identified 32 emphysema-associated loci at $p < 5 \times 10^{-5}$ where it is likely that colocalized GWAS and eQTL signals arise from the same causal variant. It should be noted that these are putative and not confirmed disease variants due to our use of a relaxed GWAS significance threshold and the inherent complexities of colocalization, which continues to be an area of active methodological development. For the genome-wide significant locus near *TGFB2*, multiple sources of publicly available and newly generated experimental data link a functional variant, rs1690789, to *TGFB2* expression in fibroblasts. These data suggest that naturally occurring genetic variability in TGF-β signaling plays a causal role in the development of emphysema.

The TGF-β family of proteins constitutes a set of highly conserved signaling pathways that play a key role in human development and many other cellular functions (*Huminiecki et al., 2009*; *Massagué, 2012*). With respect to the lung, TGF-β family proteins participate in normal lung development and are dysregulated in COPD, emphysema, asthma, and pulmonary fibrosis (*Verhamme et al., 2015*; *Morris et al., 2003*; *Thomas et al., 2016*). Genetic variants near TGF-β superfamily members *TGFB2* (*Castaldi et al., 2014*; *Cho et al., 2014*), *ACVR1B* (*Boueiz et al., 2019*), *LTBP4* (*Wain et al., 2017*), and *BMP6* (*Loth et al., 2014*) have been identified in GWAS for lung function and COPD, but prior to this study the region near *ACVR1B* was the only one linked to a gene in the TGF-β pathway through functional studies (*Boueiz et al., 2019*). Our findings demonstrate that the emphysema-associated variant rs1690789 is located in an active gene regulatory region in human lung fibroblasts that interacts with the promoter region of *TGFB2* and regulates *TGFB2* expression.

These analyses highlight the genetic and gene regulatory complexity of this region. Conditional association analyses identified two independent associations with moderate centrilobular emphysema near *TGFB2*, and both associations are in linkage equilibrium (i.e. low linkage disequilibrium) with the lead variant identified in a previous GWAS of severe COPD (*Cho et al., 2014*). In addition, the region containing rs1690789 has multiple interactions with other DNA regions, including the *TGFB2* promoter and other downstream regions, indicating that this is a region of active chromatin interaction in human lung fibroblasts.

While our analyses provide evidence that the emphysema-associated GWAS region downstream from *TGFB2* interacts with the promoter of *TGFB2* and regulates the expression of *TGFB2* in human primary lung fibroblasts, many important questions remain about the function of the emphysema-associated locus near *TGFB2*. First, the rs1690789 variant appears to be an eQTL for expression of *TGFB2* in fibroblasts, but it is also strongly associated with *TGFB2* expression in thyroid tissue in GTEx with an opposite direction of effect, suggesting complex and possibly context-dependent activity of this region. This is further supported by the observation that rs1690789 lies within a DNaseI peak in some but not all fibroblasts in the ENCODE and Roadmap projects, suggesting that the regulatory element in this region may be active only in certain fibroblast subsets, under certain conditions, or that the regulatory activity of this region is influenced by common (but unmeasured) genetic variation in these cells. Additional investigations are warranted to examine the context-specific function of this region. Second, our studies do not explain the function of the secondary association signal in this region, and it is also possible that both association regions may have functional effects in other cell types that contribute to COPD susceptibility. Third, it is possible that even within a single, statistically independent association peak, there may be multiple functional variants in tight linkage disequilibrium that contribute to the emphysema-related effects of this region. Future functional screening studies of this region can address this question. Finally, gene-level functional studies will be required to characterize the functional consequences of increased and decreased *TGFB2* expression on lung fibroblast function.

In summary, integrative GWAS-eQTL analysis of emphysema patterns identified 32 candidate loci with strong evidence of harboring gene regulatory variants responsible for the GWAS signal, including a locus near *TGFB2*. Functional investigation of the associated region near *TGFB2* confirmed the

presence of a functional variant, rs1690789, that likely contributes to the genetic predisposition to emphysema by regulating *TGFB2* expression in fibroblasts. This region has multiple independent association signals and an extensive pattern of chromosomal interaction, indicating that additional investigations are required to fully characterize the gene regulatory activity at this locus. In addition to the association near *TGFB2*, we identified dozens of other high confidence regions in our colocalization analysis, indicating additional functional variants that could be identified by high-throughput functional characterization approaches such as massively parallel reporter assays or CRISPR-mediated mutagenesis.

## Materials and methods

### Study subjects

#### COPDGene
The Genetic Epidemiology of COPD Study (COPDGene, NCT00608764, www.copdgene.org) is an ongoing multicenter, longitudinal study designed to investigate the genetic and epidemiologic characteristics of COPD. The protocols for subject recruitment and data collection for the COPDGene study have been previously described (*Regan et al., 2010*). At baseline, COPDGene enrolled 10,192 Non-Hispanic White (NHW) and African-American (AA) subjects at 21 centers across the United States between the ages of 45 and 80 years with a minimum of 10 pack-years smoking history. Subjects represented the full spectrum of disease severity as defined by the Global Initiative for Chronic Obstructive Lung Disease (GOLD) spirometric grading system. In addition to completing detailed questionnaires, pre- and post-bronchodilator spirometry, and volumetric computed tomography of the chest, participants provided whole blood for DNA genotyping.

Genotyping was performed by Illumina (San Diego, CA) on the HumanOmniExpress array. Subjects were excluded for missingness, heterozygosity, chromosomal aberrations, gender check, population outliers, and cryptic relatedness. Genotyping at the Z and S alleles was performed in all subjects. Subjects known or found to have alpha-1 antitrypsin deficiency were excluded. Markers were excluded based on missingness, Hardy- Weinberg P-values, and low minor allele frequency. Imputation on the COPDGene cohorts was performed using MaCH and minimac (version 2012-10-09) (*Li et al., 2010*; *Howie et al., 2012*). Reference panels for the non- Hispanic whites and African-Americans were the 1000 Genomes3 Phase I v3 European (EUR) and cosmopolitan reference panels, respectively.

#### ECLIPSE
The Evaluation of COPD Longitudinally to Identify Predictive Surrogate Endpoints Study (ECLIPSE; SCO104960, NCT00292552, www.eclipse-copd.com) is a longitudinal study with three-year follow-up data available for 2501 smoking subjects (2164 subjects with COPD and 337 smoking controls). The detailed study protocol and inclusion criteria have been previously published (*Vestbo et al., 2008*). For this analysis, 1519 subjects with COPD (defined as GOLD spirometric stages 2–4) and available CT scans were analyzed. COPD was defined by $FEV_1$ <80% of predicted and $FEV_1$/ FVC < 0.7.

Genotyping was performed using the Illumina HumanHap 550 V3 (Illumina, San Diego, CA). Subjects and markers with a call rate of <95% were excluded. Subjects with alpha-1 antitrypsin deficiency based on serum protein levels were excluded from this analysis. Population stratification and genotype imputation was performed using the same procedures and software as described above for COPDGene. GWAS models were adjusted for age, gender, pack-years of smoking history, and genetic ancestry via principal components.

### Common variant genetic association analysis of LHE measures
We performed GWAS analyses of the 5 LHE measurements separately in the three cohorts (COPDGene NHW, COPDGene AAs, and ECLIPSE, total N = 11,282 subjects, 18,383,174 SNPs imputed to the 1000 Genomes reference panel, version 3, hg19). Analysis was limited to imputed SNPs with an imputation $r^2$ >0.3. Imputed genotypes were analyzed using the –dosage command in PLINK v1.9 (*Chang et al., 2015*), though for SNPs with genotyped data the observed genotypes were used. GWAS models were adjusted for age, gender, pack-years of smoking history, and genetic ancestry

via principal components (*Price et al., 2006*). Results were meta-analyzed using the METAL (*Willer et al., 2010*) program using fixed effects meta-analysis with inverse variance weighting using SNP effect sizes and standard errors. We analyzed SNPs with a MAF $\geq$1%, and we meta-analyzed SNPs with results in at least two of the three cohorts.

## CT scan acquisition and generation of LHE measures

The generation of LHE measures in COPDGene has been previously described (*Castaldi et al., 2013*). For the current studies, additional LHE measures were generated in ECLIPSE CT scans using the same method.

## Clinical associations of LHE measures in ECLIPSE

LHE measurements have been previously associated with key COPD-related measures (e.g. spirometry, MMRC) (*Castaldi et al., 2013*). To test if this relationship was consistent in the measurements generated in ECLIPSE, we visualized the median percentage of each emphysema pattern by GOLD stage.

## GWAS lookups of LHE significant variants in GWAS of $FEV_1$, $FEV_1/FVC$, COPD, and smoking behavior

For the 14 lead SNPs associated with one or more of the LHE phenotypes at genome-wide significance, we queried other COPD-related GWAS for these variants or variants in linkage disequilibrium with these variants ($r^2$ >0.8 in the 1000 Genomes EUR reference panel). The queried GWAS studies were published studies of $FEV_1$ and $FEV_1/FVC$ (*Shrine et al., 2019*), COPD status (*Sakornsakolpat et al., 2019*), or history of smoking. The smoking GWAS results were obtained from the UK Biobank Pheweb server (http://pheweb.sph.umich.edu:5000/) on July 7, 2019 for the phenotype '20116_1: Smoking status: Previous.'

## eQTL data and colocalization analysis

For colocalization and cell type enrichment analyses, GWAS SNPs significant at $p<5\times10^{-5}$ were considered. GTEx version six full results for 44 tissues were downloaded from the GTEx portal (https://www.gtexportal.org/home/datasets), and eQTLs were calculated from blood RNAseq data in 385 NHW subjects from the COPDGene study using the same methods used in the GTEx Study v6 analysis. Details on the generation of COPDGene RNAseq data have been previously described (*Parker et al., 2017*). GWAS-eQTL integrative analysis was performed according to the approach previously described in *Castaldi et al. (2015)*. Briefly, for each set of eQTL results, SNPs with a significant *cis* eQTL association at a 10% FDR threshold were extracted from each of the five sets of LHE GWAS results. Q-values were calculated for each subset of GWAS SNPs separately using the q-value package (*Storey et al., 2019*), and SNPs demonstrating both significant eQTL and GWAS associations were retained for subsequent analysis (i.e. eQTL-GWAS SNPs). Within each set of eQTL-GWAS SNPs, association regions for colocalization were defined by selecting all SNPs within 250 kilobases (kb) of each independent GWAS association. Colocalization of the GWAS and eQTL signals in these regions was calculated using the Bayesian colocalization method implemented in the R package coloc (*Giambartolomei et al., 2014*) using the default settings for the prior probability of a SNP being associated to target gene expression, the GWAS phenotype, and both measures (prior probability $1 \times 10^{-4}$, $1 \times 10^{-4}$, and $1 \times 10^{-5}$, respectively).

To confirm the colocalization results for *TGFB2*, colocalization was also performed for the GWAS results for moderate centrilobular emphysema using the Sherlock method (*He et al., 2013*). This analysis was performed using all the moderate centrilobular GWAS results referenced against three GTEx v6 eQTL datasets (transformed fibroblasts, lung, and whole blood). The following parameter settings were used: cis eQTL significance threshold $p<0.001$, trans eQTL significance threshold $p<1\times10^{-5}$.

## Causal SNP estimation with PICS

To narrow the list of putative causal variants for the primary association near TGFB2, we used the probabilistic inference of causal SNPs algorithm (PICS) (*Farh et al., 2015*) which infers per SNP causal probabilities from the strength of association of the lead SNP and linkage disequilibrium

information from 1000 Genomes reference populations. The EUR reference population was used for this analysis, which was conducted via the PICS web interface (https://pubs.broadinstitute.org/pubs/finemapping/pics.php).

## Identification of variants predicted to effect transcription factor occupancy

For SNPs with a PICS causal probability of 5% or greater, we queried these SNPs against their Contextual Analysis of Transcription Factor Occupancy (CATO) model predictions (*Maurano et al., 2015*), which was trained on deep DNaseI sequencing data from the Roadmap project to predict per-SNP effects on transcription factor occupancy based on the predicted effects of each SNP on the binding energy of overlapping TF motifs and a number of factors related to local genomic sequence content. SNPs exceeding a CATO score of 0.1 were considered likely to alter TF occupancy.

## Cell type and cell line GWAS enrichment analysis with garfield

To determine whether LHE GWAS association were enriched in gene regulatory annotations from ENCODE and Roadmap Epigenomics data, we performed enrichment analysis for the LHE phenotypes with genome-wide significant results using the Garfield program and its pre-processed epigenomic annotations (*Iotchkova et al., 2019*). The GWAS significance threshold was set at $p < 5 \times 10^{-5}$, and the default parameters were used for LD pruning ($r^2 > 0.1$), LD proxy threshold ($r^2 > 0.8$), minor allele frequency binning (five bins), LD tag binning (five bins), and TSS distance binning (five bins). The significance threshold was set at $p < 0.0001$ corresponding to Bonferroni adjustment for the effective number of independent annotations.

## Overlap of rs1690789 with cell-specific DNaseI peaks

Imputed DNaseI hypersensitivity peaks from Roadmap Epigenomics cell types or cell lines (*Ernst and Kellis, 2015*) were downloaded from http://egg2.wustl.edu/roadmap/data/byFileType/peaks/consolidatedImputed/narrowPeak/. The overlap of rs1690789 with DNaseI peaks and enhancer marks was identified using the GoShifter program (*Trynka et al., 2015*), and the raw DNaseI data for these cell types was visualized using the UCSC Genome browser.

## Cell culture

IMR-90 fibroblasts were purchased from ATCC and cultured in Eagle's Minimal Essential Medium (EMEM, #12–611F, Lonza) supplemented with 10% fetal bovine serum, penicillin and streptomycin. The cells tested negative for mycoplasma by MycoAlert Detection Kit (#LT07-418, Lonza). Primary human lung fibroblast cells were isolated from the lung tissue of healthy individuals (Marsico Lung Institute, University of North Carolina at Chapel Hill, North Carolina) as previously described (*Fulcher et al., 2005*). Briefly, lung tissue samples were cut into small pieces and seeded onto culture dishes supplemented with DMEM/F12 medium, 10% fetal bovine serum, penicillin, streptomycin, amphotericin B and gentamicin. Amphotericin B and gentamicin were removed from the medium after the cells were passaged. The primary human lung fibroblasts were passaged twice and grown to 90% confluence prior to subsequent experiments. Human lung tissue was obtained under protocol #03–1396 approved by the University of North Carolina at Chapel Hill Biomedical Institutional Review Board.

## 4C data in IMR90 cell lines

4C chromosome conformation interaction results from the paper by *Rao et al. (2014)* were queried from the Yue Lab public website (http://promoter.bx.psu.edu/) using the following search parameters: Species = human, Assembly = hg19, Tissue = IMR90, Type = Lieberman VC-norm, Resolution = 10 kb, SNP = rs1690789, Extended Region = 500 kb.

## Chromatin conformation capture assay (3C)-PCR

Human lung fibroblasts IMR90 cells were cultured to 80% confluency then cross-linked and lysed followed by digestion with BglII overnight. DNA fragments were then ligated with T4 ligase (New England Biolabs, #M0202L) for 6 hr at 16°C. After purification, 3C templates were used in PCR detection

with unidirectional primers to indicate specific chromatin interaction by comparing relative band intensity from targeted regions against negative and positive control regions with three technical replicates (i.e. same 3C templates, multiple PCR repeats). Primer sequences used for 3C-PCR are listed in *Supplementary file 1* Table 10. Detailed description of our methods has been published previously (*Zhou et al., 2012*).

## CRISPR/Cas9 rs1690789 knockout

To generate the rs1690789 CRISPR/Cas9 regional knockout primary human lung fibroblast cells, two guide RNAs (u1 forward: 5'- GATACTCCAGTACATTGAGAAGG-3'; u2 forward: 5'-TGGAGTATCATTTCAGTGTTAGG-3') located upstream from the SNP and two guide RNAs (d1 forward: 5'-CAG-CAGCGAGTTTGGCACTCAGG-3'; d2 forward: 5'-TGTCTCATTGCACACTCATGGGG-3') located downstream from the SNP were cloned into pSpCas9 (BB)−2A-Puro (PX459) V2.0 vectors (Addgene plasmids #62988), individually. Plasmids were verified by DNA sequencing. FuGENEHD was applied to transfect three pairs of gRNA plasmids (u1 and d1, u1 and d2, u2 and d2) into primary normal human lung fibroblast (NHLF) cells according to the manufacturer's instructions. PX459 empty vectors were transfected as control. Forty-eight hours after transfection, cells were selected with 1.2 µg/mL puromycin. After 2–3 weeks of recovery and expansion, cells were collected for DNA, RNA extraction and qPCR. Four biological replicates were performed (i.e. same donor, four different transfections).

## Assessment of CRISPR/Cas9 editing efficiency

DNA samples from human lung fibroblast cells were extracted using QuickExtract DNA Extraction solution (#QE0905T, Lucigen, WI) following manufacturer's instructions. SYBRGreen dye-based quantitative RT-PCR was performed using the same equipment system and analysis method mentioned above, with the following primers to assess editing efficiency (forward: 5'- GTTACCGATGCTTAAATGCCAC-3'; reverse: 5'- AGAATATCCCCATGAGTGTGC-3'). The control was cells transfected with PX459 empty vector.

## Gene expression measurements by RT-PCR

Human lung fibroblast cell RNA was extracted using RNeasy Mini Kit (#74106, Qiagen, MD), and reverse transcription was performed by using High-Capacity cDNA Reverse Transcription Kit (#4374966, Applied Biosystems, MA). Quantitative RT-PCR was performed on QuantStudio 12K Flex Real-Time PCR System (Applied Biosystems) with gene-specific TaqMan probes (Hs.PT.58.24824921) from IDT (Integrated DNA technologies, IA) for detecting *TGFB2* expression. Relative gene expression level was calculated based on the standard $2^{-\Delta\Delta CT}$ method, using *GAPDH* as a reference gene. For both the TGFB2 expression and editing efficiency tests, qPCR values were normalized against the mean qPCR value for the control cells for each experiment. Comparisons were performed using unpaired t-tests.

## Study approval

Written, informed consent was obtained for all participants, and all study and consent forms were approved by the institutional review boards of the participating institutions.

## Acknowledgements

This work was supported by NHLBI U01HL089897, R01HL089897, R01HL089856, R01HL124233, R01HL126596, R01HL113264, P01105339, and P01HL114501. The COPDGene study (NCT00608764) is also supported by the COPD Foundation through contributions made to an Industry Advisory Board comprised of AstraZeneca, Boehringer Ingelheim, GlaxoSmithKline, Novartis, Pfizer, Siemens and Sunovion. The Norway GenKOLS (Genetics of Chronic Obstructive Lung Disease, GSK code RES11080) and the ECLIPSE studies (NCT00292552; GSK code SCO104960) were funded by GSK. The Marsico Lung Institute is supported by the Cystic Fibrosis Foundation (BOUCHE15R0) and NIH (DK065988). The content is solely the responsibility of the authors and does not necessarily represent the official views of the National Heart, Lung, and Blood Institute or the National Institutes of Health.

# Additional information

## Competing interests

Michael H Cho: reports grants from GSK and personal fees from Genentech. Craig P Hersh: reports personal fees from Mylan, personal fees from AstraZeneca, Concert Pharmaceuticals, 23andMe, grants from Novartis, and Boehringer-Ingelheim. George Washko: reports grants and other support from Boehringer Ingelheim, PulmonX, BTG Interventional Medicine, Janssen Pharmaceuticals and GSK. Scott H Randell: reports receiving personal fees from Amgen. Edwin K Silverman: received honoraria from Novartis for Continuing Medical Education Seminars and grant and travel support from GlaxoSmithKline. Raúl San José Estépar: reports personal fees from Boehringer Ingelheim, Eolo Medical and Toshiba. Peter J Castaldi: has received research support and consulting fees from GSK and Novartis. The other authors declare that no competing interests exist.

## Funding

| Funder | Grant reference number | Author |
| --- | --- | --- |
| National Heart, Lung, and Blood Institute | R01 HL124233 | Peter J Castaldi |
| National Heart, Lung, and Blood Institute | R01 HL126596 | Peter J Castaldi |
| NHLBI | R01HL089897 | Edwin K Silverman |
| NHLBI | R01HL089856 | James Crapo |
| NHLBI | R01HL113264 | Edwin K Silverman |
| NHLBI | P01105339 | Edwin K Silverman |
| NHLBI | P01HL114501 | Edwin K Silverman |

The funders had no role in study design, data collection and interpretation, or the decision to submit the work for publication.

## Author contributions

Margaret M Parker, Formal analysis, Writing—original draft, Writing—review and editing; Yuan Hao, Feng Guo, Formal analysis, Investigation, Writing—original draft, Writing—review and editing; Betty Pham, Formal analysis, Investigation, Writing—review and editing; Robert Chase, Data curation, Formal analysis, Writing—review and editing; John Platig, Victor J Thannickal, Writing—review and editing; Michael H Cho, Data curation, Writing—review and editing; Craig P Hersh, Funding acquisition, Project administration, Writing—review and editing; James Crapo, Edwin K Silverman, Funding acquisition, Writing—review and editing; George Washko, Data curation, Funding acquisition, Writing—review and editing; Scott H Randell, Resources, Data curation, Funding acquisition, Writing—review and editing; Raúl San José Estépar, Conceptualization, Data curation, Formal analysis, Methodology, Writing—review and editing; Xiaobo Zhou, Conceptualization, Formal analysis, Supervision, Investigation, Writing—original draft, Project administration, Writing—review and editing; Peter J Castaldi, Conceptualization, Formal analysis, Funding acquisition, Investigation, Methodology, Writing—original draft, Project administration, Writing—review and editing

## Author ORCIDs

Peter J Castaldi (iD) https://orcid.org/0000-0001-9920-4713

## Decision letter and Author response

Decision letter https://doi.org/10.7554/eLife.42720.022
Author response https://doi.org/10.7554/eLife.42720.023

## Additional files

### Supplementary files
• Supplementary file 1. This file contains Tables 1-11.
DOI: https://doi.org/10.7554/eLife.42720.013
• Transparent reporting form
DOI: https://doi.org/10.7554/eLife.42720.014

### Data availability
COPDGene genetic data and RNA-seq data have been deposited in dbGaP under accession code phs000765.v3.p2. To access these data users may apply for access to the dbGaP data repository (https://www.ncbi.nlm.nih.gov/books/NBK482114/).

The following dataset was generated:

| Author(s) | Year | Dataset title | Dataset URL | Database and Identifier |
|---|---|---|---|---|
| Parker MM, Chase RP, Lamb A, Reyes A, Saferali A, Yun JH, Himes BE, Silverman EK, Hersh CP, Castaldi PJ | 2017 | Blood RNA-seq | https://www.ncbi.nlm.nih.gov/projects/gap/cgi-bin/study.cgi?study_id=phs000765.v3.p2 | NCBI dbGaP, phs000765.v3.p2 |

The following previously published datasets were used:

| Author(s) | Year | Dataset title | Dataset URL | Database and Identifier |
|---|---|---|---|---|
| GTEx Consortium | 2015 | GTEx | https://storage.google-apis.com/gtex_analysis_v6/single_tissue_eqtl_data/GTEx_Analysis_V6_eQTLs.tar.gz | GTEx Portal, GTEx v6 |
| ENCODE | 2012 | ENCODE | https://egg2.wustl.edu/roadmap/data/byFile-Type/peaks/consolidatedImputed/narrowPeak/ | ENCODE and ROADMAP, narrowPeak |

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
