## [Decision Letter]

Thank you for submitting your article "Identification of an emphysema-associated genetic variant near *TGFB2* with regulatory effects in lung fibroblasts" for consideration by *eLife*. Your article has been reviewed by three peer reviewers, including Andrew P Morris as the Reviewing Editor and Reviewer #1, and the evaluation has been overseen by Mark McCarthy as the Senior Editor. The following individual involved in review of your submission has agreed to reveal their identity: Louise Wain (Reviewer #2).

The reviewers have discussed the reviews with one another and the Reviewing Editor has drafted this decision to help you prepare a revised submission.

Summary:

The reviewers appreciated the work in bringing together the largest GWAS of emphysema to date, and agreed that the integration of GWAS findings with eQTL resources and additional epigenomic data was a good demonstration of how to move from locus to causal variant and effector gene. However, the reviewers were concerned that the association signal at the TGFB2 locus was driven by just COPDGene NHW, with no evidence of association in COPDGene AA or ECLIPSE.

Essential revisions:

1) Independent replication of the emphysema association signal at the TGFB2 locus is *essential*. Ideally, this would be in an additional study of emphysema, although supporting evidence from related phenotypes (e.g. COPD or lung function) would be an alternative.

2) More details are needed for the colocalization part of the work. As of now, only the reference for the approach is noted. It would be helpful to know the thresholds used and the results they observed. In addition, there are many different colocalization approaches and they tend to not always agree. We ask that the authors assess whether the results are robust to different colocalization methods.

3) The provision of the full results of the eQTL colocalisation analysis is commendable and potentially useful resource for the community. However, the authors should provide the caveat that at P<5x10^-5^ there are likely to be many false positive emphysema GWAS associations and so replication of the GWAS results should be sought prior to embarking on pursuit of the genes implicated by this analysis.

4) The associations on chromosome 15 (CHRNA3/f locus) and chromosome 19 (CYP2A6) reflect an effect on lung function via smoking behaviour and thus point primarily to addiction pathways. Accepting that it is difficult to entirely adjust for smoking in these analyses, the authors could comment on this and perhaps report (through comparison with published smoking GWAS data) which of the signals might be driven by smoking.

5) The association with the SERPINA1 z-allele is of interest but suggests that there might be individuals with alpha-1 anti-tryspin deficiency amongst the cases. This information should be provided.

6) The authors mentioned that they used PICS to derive "likelihoods" that variants are causal. It would be useful to have a brief description of this approach in the Materials and methods. What is the likelihood of the chosen SNP versus the other six? What is the motivation for the 5% threshold? A more usual approach would be to build a 99% credible set, and then interrogate those variants instead. Posterior probabilities should be provided for variants considered in downstream interrogation

7) For enrichments, the authors use 125 cell types from the Roadmap Epigenomics segmentations (subsection “Overlap of LHE SNPs with Epigenomic Marks in Roadmap Epigenomics Cell Types”) but then state that enhancer were defined by collapsing states 13-18 from Ernst and Kellis, 2015. However, Ernst and Kellis, 2015, does not describe the Roadmap Epigenomics chromatin states, so the reviewers were confused about how the enhancer states were actually generated.

8) The GoShifter approach has been shown to be suboptimal for calculating enrichments (Iotchkova et al., 2019) and we would instead recommend running GARFIELD, which has built-in Roadmap and ENCODE annotations. This would strengthen the enrichment results.

9) In the Discussion the authors note that the underlying regulatory element is called in some but not all fibroblasts data sets and hypothesize that this could represent restricted activity to some subset of fibroblasts, or some subset of conditions. If this is truly the causal variant, then couldn't the element call be a function of sample genotype too? And, thus if the ENCODE/Roadmap/etc. sample did not have the "active" genotype, a peak call might not be observable. We would encourage the authors to add this as a possible interpretation, unless there is justified motivation to report otherwise.

---

## [Author Response]

Essential revisions:1) Independent replication of the emphysema association signal at the TGFB2 locus is essential. Ideally, this would be in an additional study of emphysema, although supporting evidence from related phenotypes (e.g. COPD or lung function) would be an alternative.

We have included new results where we have reported the significant and direction of effect for all of our genome-wide significant associations in relation to FEV1, FEV1/FVC, and COPD status. These results were queried from the two largest GWAS meta-analyses by Shrine (spirometry measures) and Sakornsakolpat (COPD) which were both published this year in Nature Genetics (Table 4 in Supplementary file 1, Introduction, and subsection “GWAS Lookups of LHE Significant Variants in GWAS of FEV1, FEV1/FVC, COPD, and Smoking Behavior”).

2) More details are needed for the colocalization part of the work. As of now, only the reference for the approach is noted. It would be helpful to know the thresholds used and the results they observed. In addition, there are many different colocalization approaches and they tend to not always agree. We ask that the authors assess whether the results are robust to different colocalization methods.

We have clarified in the Materials and methods section the relevant parameters used for the colocalization analysis (subsection “eQTL Data and Colocalization Analysis”). To address the issue of the robustness of key colocalization results, we performed a separate colocalization analysis for the moderate centrilobular GWAS associations using the Sherlock method (Table 8 in Supplementary file 1, subsection “Validation of LHE Clinical and Genetic Associations”, and, subsection “Identification of Variants Predicted to Effect Transcription Factor Occupancy”). This analysis also identified TGFB2 as a colocalizing gene for this phenotype in GTEx fibroblasts expression data. We agree that colocalization results can often be parameter and method dependent and that functional validation of colocalization results is necessary. We have added text to the Discussion to make this point clear.

3) The provision of the full results of the eQTL colocalisation analysis is commendable and potentially useful resource for the community. However, the authors should provide the caveat that at P<5x10^-5^ there are likely to be many false positive emphysema GWAS associations and so replication of the GWAS results should be sought prior to embarking on pursuit of the genes implicated by this analysis.

We agree and this caveat has been added to the Discussion.

4) The associations on chromosome 15 (CHRNA3/f locus) and chromosome 19 (CYP2A6) reflect an effect on lung function via smoking behaviour and thus point primarily to addiction pathways. Accepting that it is difficult to entirely adjust for smoking in these analyses, the authors could comment on this and perhaps report (through comparison with published smoking GWAS data) which of the signals might be driven by smoking.

We have collected the p-values for association to smoking status for our genome-wide significant variants from the UK Biobank Pheweb server (Table 5 in Supplementary file 1, subsection “GWAS Lookups of LHE Significant Variants in GWAS of FEV1, FEV1/FVC, COPD, and Smoking Behavior”). Two of the 10 loci show association to smoking status at the 15q25 and 19q13 loci (subsection “Validation of LHE Clinical and Genetic Associations”, last paragraph).

5) The association with the SERPINA1 z-allele is of interest but suggests that there might be individuals with alpha-1 anti-tryspin deficiency amongst the cases. This information should be provided.

Subjects with alpha-1 antitrypsin deficiency were excluded from both COPDGene and ECLIPSE by genotyping and serum protein levels, respectively. However, we then checked the imputed genotype classes in ECLIPSE, and we identified six subjects imputed to have the PiZZ genotype. We accordingly excluded these subjects from the ECLIPSE analyses, which resulted in a higher p-value (0.003 versus 0.0004) with a consistent direction of effect. We repeated the meta-analysis (new p-value 1x10^-7^) and included these results in the text (subsection “Validation of LHE Clinical and Genetic Associations”, Materials and methods subsection “ECLIPSE” and “Common Variant Genetic Association Analysis of LHE Measures”).

6) The authors mentioned that they used PICS to derive "likelihoods" that variants are causal. It would be useful to have a brief description of this approach in the Materials and methods. What is the likelihood of the chosen SNP versus the other six? What is the motivation for the 5% threshold? A more usual approach would be to build a 99% credible set, and then interrogate those variants instead. Posterior probabilities should be provided for variants considered in downstream interrogation

We had inadvertently omitted the description of the methods for PICS variant prioritization and the use of the previously published model by Maurano et al. for predicting SNPs likely to cause allelic imbalance. These have now been included (subsection “Identification of Variants Predicted to Effect Transcription Factor Occupancy”), and we have updated the text (subsection “Fine Mapping Identifies a Candidate Causal Variant in the TGFB2 Locus”) to clarify our approach, which is as follows: we wished to use multiple methods to produce a small a set of putative functional variants as possible for functional prioritization. In this case, rs1690789 was the only variant that had a reasonable PICS likelihood (i.e. > 5% causal probability) that was also predicted to cause allelic imbalance based on the Maurano model. Fortunately, this aggressive bioinformatic prioritization seems to have been successful for identifying a functional region near TGFB2, but we have taken care to state in the Discussion that there may well be other disease causing functional regions near TGFB2 that would be identified through a more comprehensive (and expensive) functional screening approach, such as MPRA.

7) For enrichments, the authors use 125 cell types from the Roadmap Epigenomics segmentations (subsection “Overlap of LHE SNPs with Epigenomic Marks in Roadmap Epigenomics Cell Types”) but then state that enhancer were defined by collapsing states 13-18 from Ernst and Kellis, 2015. However, Ernst and Kellis, 2015, does not describe the Roadmap Epigenomics chromatin states, so the reviewers were confused about how the enhancer states were actually generated.

Ernst and Kellis, 2015, refers to the development of ChromImpute epigenomic marks from Roadmap experiments, which are the marks that we used to identify overlap between rs1690789 and DNaseI peaks in Roadmap cell lines and cell types (subsection “CRISPR/Cas9 rs1690789 knockout”). We have removed reference to the enhancer states (which were generated by collapsing Chromimpute states 13-18), since our analysis of rs1690789 overlap is limited to DNaseI peaks.

8) The GoShifter approach has been shown to be suboptimal for calculating enrichments (Iotchkova et al., 2019) and we would instead recommend running GARFIELD, which has built-in Roadmap and ENCODE annotations. This would strengthen the enrichment results.

We completed the GARFIELD enrichment analysis and have included these results in the manuscript (subsection “eQTL Colocalization Analysis to Identify Candidate GWAS Target Genes and Tissue Enrichment of LHE GWAS signals”, last paragraph, Materials and methods subsection “Overlap of rs1690789 with Cell-Specific DNaseI Peaks”), which confirm enrichment of moderate centrilobular GWAS signals in fibroblast DNaseI peaks. We appreciate this very helpful suggestion.

9) In the Discussion the authors note that the underlying regulatory element is called in some but not all fibroblasts data sets and hypothesize that this could represent restricted activity to some subset of fibroblasts, or some subset of conditions. If this is truly the causal variant, then couldn't the element call be a function of sample genotype too? And, thus if the ENCODE/Roadmap/etc. sample did not have the "active" genotype, a peak call might not be observable. We would encourage the authors to add this as a possible interpretation, unless there is justified motivation to report otherwise.

We agree with this point and have included this in the text (Help “eQTL Colocalization Analysis to Identify Candidate GWAS Target Genes and Tissue Enrichment of LHE GWAS signals” third paragraph and Discussion, fourth paragraph) as a potential explanation for variability in epigenetic marks across fibroblast cell types in Roadmap.